# Hydrothermal Sphalerites from Ore Deposits of Baia Mare Area

**Gheorghe Damian** [1,*], **Andrei Buzatu** [1], **Andrei Ionuţ Apopei** [1], **Floarea Damian** [2] and **Andreea Elena Maftei** [3]

[1] Department of Geology, Faculty of Geography and Geology, "Alexandru Ioan Cuza" University of Iași, 700505 Iași, Romania; andrei.buzatu@uaic.ro (A.B.); andrei.apopei@uaic.ro (A.I.A.)

[2] North University Centre of Baia Mare, Technical University of Cluj-Napoca, 430083 Baia Mare, Romania; floareadamian@yahoo.com

[3] Institute for Interdisciplinary Research, Department of Exact and Natural Sciences, "Alexandru Ioan Cuza" University of Iași, 700057 Iași, Romania; andreea.maftei@uaic.ro

* Correspondence: gheorghe.damian@uaic.ro

**Abstract:** Sphalerite is an abundant mineral in the hydrothermal deposits from the Baia Mare and Oaș areas (northwestern Romania). Sphalerite samples were analyzed with an electron probe microanalyzer and Raman spectroscopy. The obtained results indicated different amounts of Fe in the various deposits from the Baia Mare and Oaș areas. The sphalerites from Baia Sprie, Cavnic, Iba, Turț Penigher, and Breiner have a low Fe wt.% content. High Fe wt.% contents are at Herja and partly at Ghezuri and Nistru (copper stage) where sphalerite is associated with pyrrhotite. The correlation between iron and zinc from sphalerites is strongly negative. The negative correlation shows that iron is the main element that replaces zinc in the sphalerite structure. The manganese content of sphalerites in the Baia Mare and Oaș area is up to 0.84 wt.%. The cadmium content is quite uniform in the Baia Mare and Oaș area with contents ranging from 0.01 to 0.72 wt.%. The Fe content of sphalerites is an important indicator of the physico-chemical conditions of deposit formation because it is a function of temperature, pressure, and sulfur fugacity.

**Keywords:** sphalerite; hydrothermal; ore deposits; Baia Mare; trace elements

## 1. Introduction

Sphalerite is a common mineral in base-metal hydrothermal mineralizations in the metallogenetic district of Baia Mare and Oaş [1–3] with very high economic importance.

Experimental studies on the importance of sphalerite geothermometers were performed by the authors of [4]. The determination of the sphalerite deposition temperature was tested by quantifying the iron content of the sphalerite sample. The research in [4] could not be confirmed by [5,6], respectively, because sphalerite in equilibrium with pyrrhotite and pyrite contains increasing amounts of FeS as the temperature drops below 742 °C. The authors of [7–9] discussed the possibility of using sphalerite as a geobarometer, not as a geothermometer.

Sphalerite is an important mineral host for many minor and trace elements. Studies using laser ablation inductively coupled mass spectroscopy (LA-ICPMS) techniques to investigate the distribution of minor and trace elements in hydrothermal sphalerite were made in [10]. The minor and trace elements have been included in sphalerite by $Zn^{2+}$ substitution [9].

Studies using LA-ICPMS to investigate the distribution of minor elements in hydrothermal sphalerites were performed in [10]. The chemical composition of sphalerite is a good indicator for the mechanism of hydrothermal ore deposits formed [11].

The aim of this study is to analyze the contents of iron and other minor elements in sphalerite to show the conditions of formation. Raman spectroscopy investigations aimed to determine the influences of the elements on the spectra of sphalerite. Raman spectra were also used to measure iron concentrations in hydrothermal sphalerite.

## 2. Geological and Petrogenesis Data

The Baia Mare and Oaș districts represent the NW part of the Neogene volcanic chain inside the eastern Carpathian Mountains and are the product of the collision between the African and European plates [11,12].

The pre-Neogene basement including crystalline rocks belonging to the "Median Dacides" and Cretacic-Paleogene formations [12] have been deeply displaced by pre-Neogene latitudinal fractures (E–W), that controlled the development of volcanism and metallogenesis [13]. These E–W strike fractures had a releaser and then localization role, especially in the adjacent zones along the support fractures of ore deposits [14]. In the north part of the Dragoș–Vodă fracture, the volcanic activity is predominant and at the south, in Țibleș–Rodna, the activity is intrusive magmatic [15].

In the Oaș–Gutâi Mountains, the magmatic activity began in the Lower Badenian [16] about ~14.8–15.1 Ma [17] and continues with the second (Sarmatian–Pontian) and the third cycle (Upper Pliocene) [18–20]. According to the present configuration from the Gutâi Mts., the magmatic rocks migrated from west to east, the oldest rocks being in the west and the younger rocks in the east of the volcanic belt (Figure 1) [18].

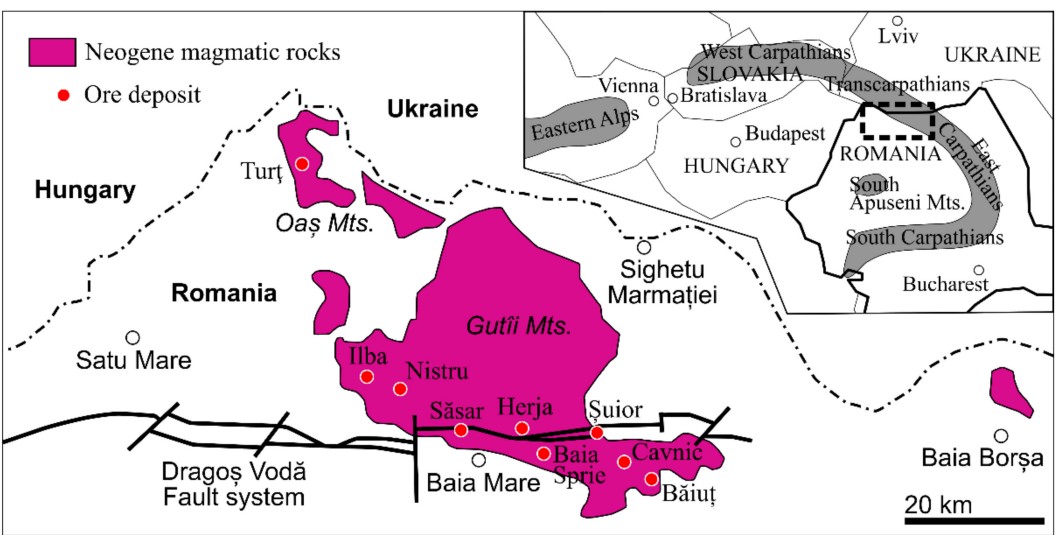

**Figure 1.** Geological sketch map of the Oaș–Gutâi Mountains, showing the location of the ore deposits in the Baia Mare metallogenetic district associated with Neogene volcanism. The inset shows a general map of the Carpathian region and location of the Baia Mare metallogenetic district (with modification after [21]).

The recent K-Ar dating [16,17] on the products of the calc–alkaline intermediary volcanism show rapid evolution between 12.0–9.5 million years in the Oaș Mts. and between 13.4–9.0 million years in the Gutâi Mts. The intermediary volcanism was restricted from the Upper Sarmatian–Pliocene to the Lower Sarmatian–Pannonian. Only a final basaltic stage in the Gutâi Mts. would belong to the Upper Pannonian–Pontian (8–6.9 million years). The K-Ar dating on the intrusions in the eastern part of the Gutâi Mts. that are not directly related to the lava flows, indicates an age of 11–9 million years. According to data presented by [18], as well as the radiometric data in the Gutâi Mts., one could note the development of volcanism from west to east and from south to north.

The intrusive magmatic activity was subsequent to the second volcanic cycle (Sarmatian–Pontian) and has an important role in the formation and placement of ore deposits [22].

### 3. Metallogeny of the Baia Mare and Oaş Area

The metallogenetic activity corresponds to three distinct phases [18]: Sarmatian, Pannonian, and Pontian. It is situated on the Southern border of the Gutâi Mts. and follows the migration of volcanism from west to east. The first stage of copper, base-metal, and gold mineralization overlays the Sarmatian pyroxene andesites in Ilba–Nistru area. The second stage corresponds to the Săsar–Valea Roşie–Crucii Hill, gold mineralization associated with the Pannonian quartz andesites. The third stage base-metal mineralization located east of the Gutâi Mts. is associated with the Pontian pyroxene andesites [18,23]. According to the authors of [15] the metallogenetic activity is predominant epithermal at the north of the Dragoş–Vodă transcrustal fault with base-metal mineralization rich in gold and silver, sometimes containing mercury, and at the south, in Ţibleş, it has a mesothermal-hypothermal quality and sometimes even a pyrometasomatic one. The metallogeny of the Oaş Mts. is similar to the third stage mineralization in the Gutâi Mts. [1] and is predominantly base metal and subordinately exhibits mercury and gold mineralizations.

Analyzing the K-Ar and Ar-Ar dating [24], the metallogenetic activity took place only in Pannonian (11.5–7.9 Ma).

According to [25] in the western part of Ilba–Nistru–Sasar the metallogenetic activity is associated with the formation of two calderas in Sarmatian and Pannonian. In the eastern part, it is associated with the Bogdan Vodă–Dragoş Vodă fault system [21,26,27]. This transcrustal fault localized the development of the volcanism and of the metallogenesis [13]. It is also possible that it had the role of a releaser and also a localizer in the surrounding spaces, along with the support fractures of the ore deposits [14].

We consider in agreement with these data that in the Baia Mare and Oas areas, metallogenesis would have taken place in a single phase in Pannonian. In this phase, there would be three successive stages of mineralization: copper, base metal, and gold–silver. These stages can be distinguished in mineralizations where there is a horizontal [28] or vertical [29] zonality. The lower part of the mineralizations is located in intrusive structures and the upper parts form large branches hosting predominantly gold-bearing stockwork.

Many mineralizations are disposed together with the intrusive bodies along some support fractures of E-W and NW-SE strike, forming tectono-magmatic-metallogenetic alignments [20,21]. These fractures were the conduits to the surface of hydrothermal solutions.

The poly-ascendent feature of the mineralizations in the Baia Mare district could be due to the change of their chemical composition besides the pulsing character of the hydrothermal solutions and to reactivation of fractures as a result of the tectonics of the magmatic arcs. The monoascendent character is predominant for the gold veins and veins of small dimensions.

### 4. Materials and Methods

The chemical compositions were determined using a Cameca SX-100 Electron Probe Microanalyzer (EPMA). The analyses were carried out at the State Geological Institute of Dionyz Stur (Bratislava, Slovakia). The analysis points were selected using the backscattered electron (BSE) images. The measurements were performed on polished carbon-coated sections using an acceleration voltage of 25 kV and a 15–20 nA beam current, 5 μm beam diameter, 20 s integration time for the peak, and 7 s for the background. The following X-ray lines from the natural (n) and synthetic (s) standards were used: S Kα (n-$CuFeS_2$), Zn Kα (n-ZnS), Fe Kα (n-$CuFeS_2$), Cu Kα (n-$CuFeS_2$), and pure metals for Mn Kα, Cd Lα, Ag Lα, and In Lα.

The Raman spectra of sphalerite samples were acquired using a Horiba Jobin–Yvon RPA–HE 532 Raman Spectrograph (Department of Geology, Alexandru Ioan Cuza University of Iaşi, Romania) with a multichannel air-cooled ($-70\,^{\circ}$C) CCD detector, using a frequency doubled Nd:YAG laser at 532 nm at a nominal power of 100 mW. The spectral range was 200–3400 cm$^{-1}$, and the spectral resolution was 3 cm$^{-1}$. The Raman system includes a "Superhead" fiber-optic Raman probe for non-contact measurements, with a 50X LWD Olympus objective used in the visible optical domain. Sulfur and cyclohexane bands were used for Raman spectra frequency calibrations. The spectra were acquired using 2 s exposures, 30 acquisitions, at a laser power of 70%, in order to improve the signal-to-noise ratio.

## 5. Results

### 5.1. Mineral Paragenesis and Geochemistry of Sphalerite

In the Baia Mare and Oaş mining districts, there are base-metal and gold–silver hydrothermal ore deposits. In the base-metal ores there are the following parageneses: pyrite–chalcopyrite–sphalerite–galena–calcite–quartz and pyrite–sphalerite–galena–quartz–calcite–barite. The most important mineral of this paragenesis is sphalerite. Representative mineral assemblages and sphalerite associations typical for several ore deposits from the Baia Mare–Oaș District are shown in Figures 2 and 3. These parageneses have formed in ores with massive texture and which were mostly deposed by boiling. Sphalerite is presented as massive aggregates while cubic or octahedral crystals with sizes of a few centimeters were rarely identified at Herja, Baia Sprie, Cavnic. This mineral can offer information about the change in the chemical composition of the mineralizing fluids [11,30].

At Ghezuri–Turţ in the Oaş Mountains, the mineralization is associated with an intrusion of laccolite (Figure 4). The mineralization is localized on the flanks of a lacolite with intense potassium altered porphyry microdiorites and has a base-metallic character.

Sphalerite has a slight tendency to enrich from the upper to the lower horizons. It appears in association with pyrite–ilvaite–sphalerite–chalcopyrite–pyrrhotite–magnetite–adularia–chlorite; pyrite–chalcopyrite–sphalerite–galena–quartz; sphalerite–galena–pyrite–siderite–kaolinite.

Macroscopically it appears as a black to brown color intergrown with galena, pyrite, chalcopyrite and is associated with kaolinite (Figure 2a). Microscopically it comes in the form of homogeneous masses or xenomorphic grains that contain inclusions of chalcopyrite (with exsolution of cubanite), pyrite, and galena.

Fe contents in sphalerite ($X_{FeS}$) range from 2.3 wt.% to 8.54 wt.% (Table 1 or Figure 5). Sphalerite from the first generation has amounts of Fe which range from 2.40 wt.% to 7.27 wt.%, the second generation from 5.5 wt.% to 5.8 wt.%, and third generation with siderite from 7.4 wt.% to 8.54 wt.%. Sphalerite from geodes has small amounts of Fe which range from 2.3 wt.% to 3.2 wt.%.

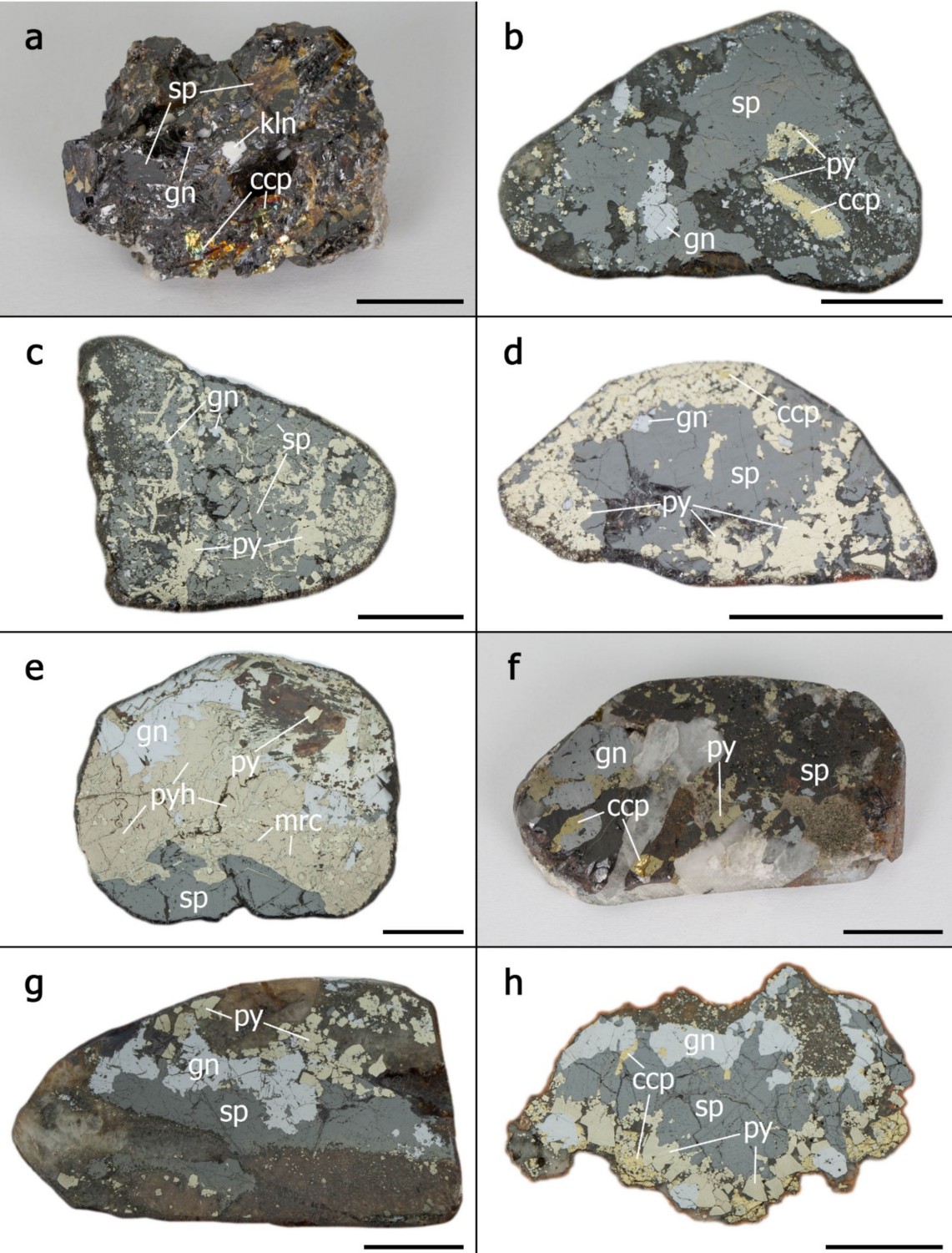

**Figure 2.** Macrophotographs of representative mineral assemblages of the polymetallic mineralization from the Baia Mare–Oaș metallogenetic district: (**a**) Ghezuri; (**b**) Ilba; (**c**) Nistru; (**d**) Săsar; (**e**) Herja; (**f**) Baia Sprie; (**g**) Cisma-Banduriţa; (**h**) Cavnic. Abbreviations: sp—sphalerite; py—pyrite; gn—galena; ccp—chalcopyrite; kln—kaolinite; mrc—marcasite; pyh—pyrrhotite. The scale bar—1 cm.

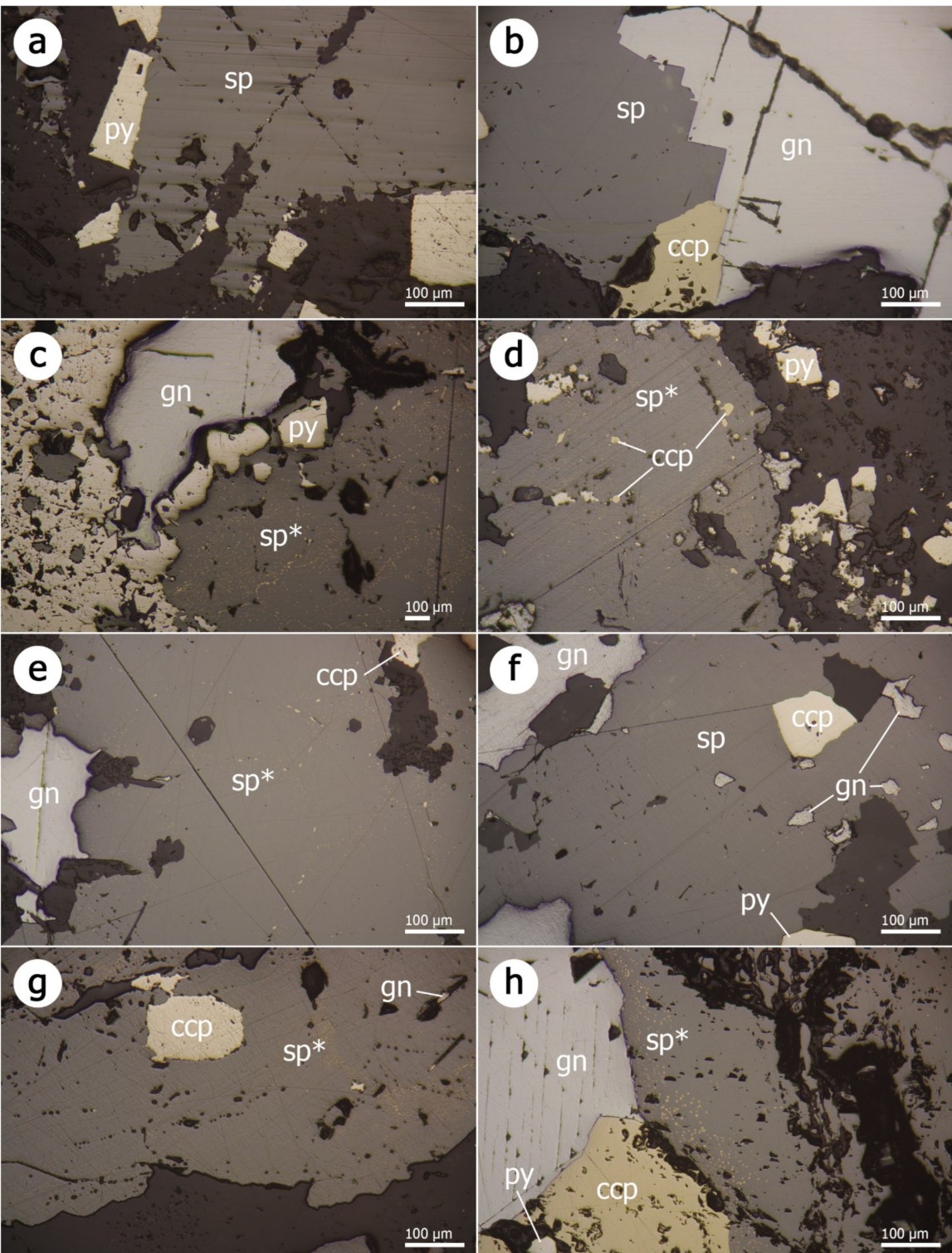

**Figure 3.** Plane-polarized reflected light photomicrographs of representative mineral assemblages of the polymetallic mineralization from the Baia Mare–Oaș metallogenetic district: (**a**) Turț-Penigher; (**b**) Ilba; (**c**) Nistru; (**d**) Dealul Crucii; (**e**) Baia Sprie; (**f**) Cisma; (**g**) Șuior; (**h**) Cavnic. Abbreviations: sp—sphalerite; sp*—sphalerite with chalcopyrite disease; py—pyrite; gn—galena; ccp—chalcopyrite.

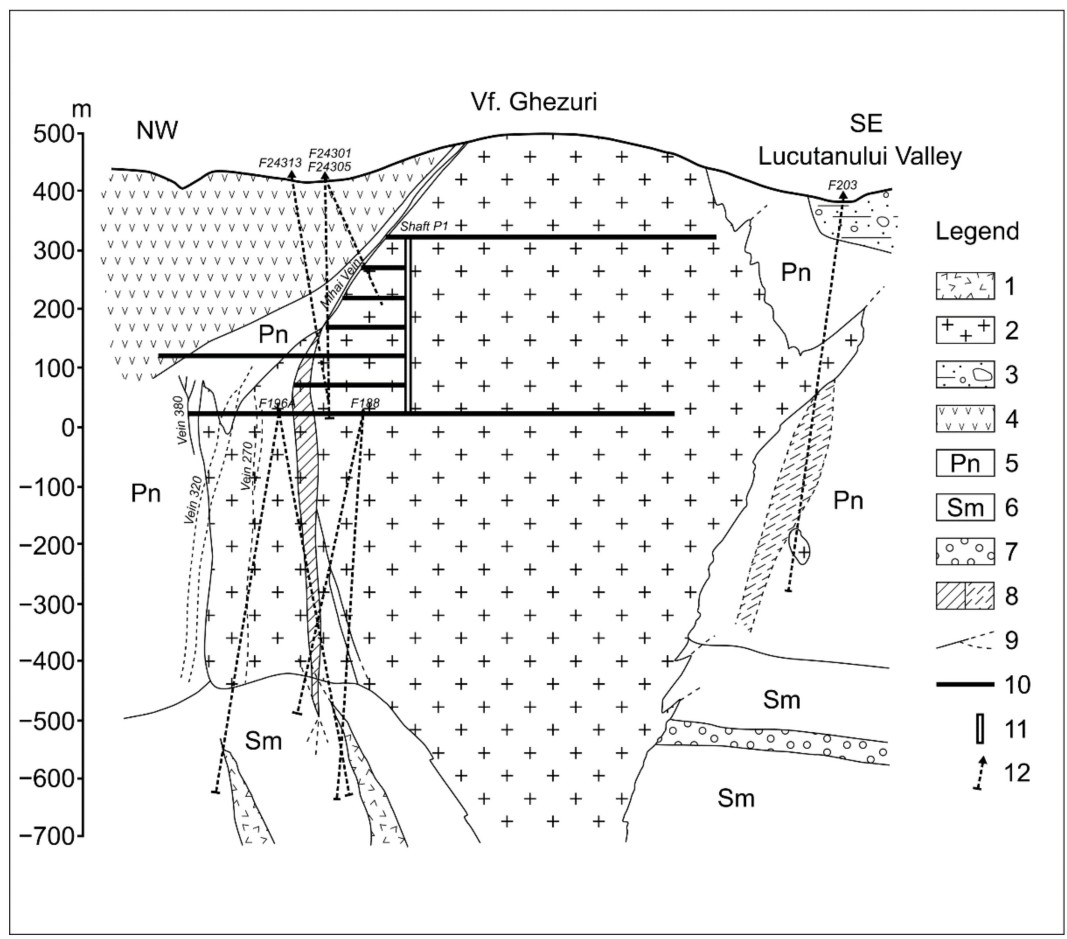

**Figure 4.** Cross-section through Turţ Ghezuri ore deposit (after [31]). 1. porphyry micro-diorite; 2. porphyry micro-granodiorite; 3. volcano-sedimentary formation; 4. pyroxene hyalodacites; 5. Pannonian (clays, sandstones, marls); 6. Sarmatian (marls, sandstones, clays); 7. Badenian (marls, sandstones, tuffs); 8. vein zone; 9. veins; 10. gallery; 11. shaft; 12. drillings.

**Table 1.** Chemical composition of sphalerite from the Baia Mare and Oaş areas.

| Sample No. | wt.% | | | | | | | | | Ore Deposits |
|---|---|---|---|---|---|---|---|---|---|---|
| | S | Zn | Fe | Mn | Cu | Cd | Ag | In | Total | |
| 12 | 35.0 | 58.87 | 5.72 | 0.20 | n.d. | n.d. | n.d. | n.d. | 99.79 | Turţ Ghezuri |
| 28 | 33.4 | 63.75 | 2.40 | 0.11 | n.d. | n.d. | n.d. | n.d. | 99.66 | Turţ Ghezuri |
| C | 31.0 | 65.60 | 3.23 | n.d. | n.d. | n.d. | n.d. | n.d. | 99.83 | Turţ Ghezuri |
| J | 33.0 | 64.34 | 2.29 | 0.19 | n.d. | n.d. | n.d. | n.d. | 99.82 | Turţ Ghezuri |
| 246 | 32.7 | 59.68 | 7.27 | 0.35 | n.d. | n.d. | n.d. | n.d. | 100 | Turţ Ghezuri |
| 260 | 35.7 | 58.85 | 4.41 | 0.04 | n.d. | n.d. | n.d. | n.d. | 99 | Turţ Ghezuri |
| 520 | 33.3 | 59.84 | 5.80 | 0.06 | n.d. | n.d. | n.d. | n.d. | 99 | Turţ Ghezuri |
| 535 | 33.5 | 61.86 | 4.57 | 0.07 | n.d. | n.d. | n.d. | n.d. | 100 | Turţ Ghezuri |
| 590 | 33.6 | 60.81 | 5.49 | 0.10 | n.d. | 0.09 | n.d. | n.d. | 100.09 | Turţ Ghezuri |
| 597 | 34.2 | 57.10 | 8.54 | 0.16 | n.d. | 0.13 | n.d. | n.d. | 100.13 | Turţ Ghezuri |
| 599 | 34.1 | 58.32 | 7.43 | 0.15 | n.d. | 0.09 | n.d. | n.d. | 100.09 | Turţ Ghezuri |
| TP-1 | 31.2 | 65.573 | 3.19 | n.d. | n.d. | 0.03 | n.d. | n.d. | 99.963 | Turt Penigher |
| TP-3 | 31.2 | 62.50 | 6.25 | n.d. | n.d. | 0.05 | n.d. | n.d. | 100 | Turt Penigher |
| Il-6 | 32.6 | 65.10 | 2.07 | 0.05 | n.d. | 0.18 | n.d. | n.d. | 100 | Ilba |
| Il-13 | 33.9 | 63.86 | 2.15 | 0.05 | 0.01 | 0.05 | n.d. | n.d. | 100.02 | Ilba |
| N-17 | 31.1 | 64.48 | 4.04 | 0.03 | 0.02 | 0.33 | n.d. | n.d. | 100 | Nistru |
| N-491 | 30.7 | 66.50 | 2.38 | 0.03 | n.d. | 0.39 | n.d. | n.d. | 100 | Nistru |
| S-8 | 32.6 | 57.32 | 9.60 | 0.48 | 0.07 | 0.03 | n.d. | n.d. | 100.1 | Săsar |

**Table 1.** *Cont.*

| Sample No. | wt.% | | | | | | | | | Ore Deposits |
|---|---|---|---|---|---|---|---|---|---|---|
| | **S** | **Zn** | **Fe** | **Mn** | **Cu** | **Cd** | **Ag** | **In** | **Total** | |
| S-11 | 32.8 | 52.17 | 8.35 | 0.59 | 6.08 | 0.01 | n.d. | n.d. | 100 | Săsar |
| S-20 | 31.8 | 59,59 | 8.38 | 0.21 | n.d. | 0.02 | n.d. | n.d. | 40.41 | Săsar |
| 1811_an 6 | 32.58 | 60.68 | 5.12 | 0.01 | 0.06 | 0.15 | n.d. | 0.01 | 98.61 | Dealul Crucii |
| 434_an 3 | 32.92 | 57.53 | 7.42 | 0.84 | 0.01 | 0.20 | 0.03 | 0.01 | 98.96 | Dealul Crucii |
| 434_an 4 | 33.53 | 60.92 | 5.16 | 0.70 | 0.09 | 0.18 | n.d. | 0.01 | 100.59 | Dealul Crucii |
| H-14 | 32.0 | 64.496 | 3.30 | 0.03 | n.d. | 0.17 | n.d. | n.d. | 99.996 | Herja |
| H-1 | 31.5 | 55.26 | 12.9 | 0.31 | n.d. | 0.03 | n.d. | n.d. | 100 | Herja |
| H-122 | 31.3 | 58.69 | 9.46 | 0.25 | n.d. | 0.30 | n.d. | n.d. | 100 | Herja |
| H-127 | 32.4 | 56.35 | 10.9 | 0.20 | n.d. | 0.15 | n.d. | n.d. | 100 | Herja |
| H-134 | 32.1 | 63.49 | 4.15 | 0.22 | n.d. | 0.04 | n.d. | n.d. | 100 | Herja |
| H-143 | 31.9 | 61.71 | 5.85 | 0.45 | 0.05 | 0.04 | n.d. | n.d. | 100 | Herja |
| H-147 | 31.7 | 60.33 | 7.86 | 0.10 | n.d. | 0.01 | n.d. | n.d. | 100 | Herja |
| H-8255 | 32.59 | 53.95 | 11.54 | 0.46 | n.d. | 0.11 | n.d. | n.d. | 98.65 | Herja |
| 6757-6 | 38.6 | 52.6 | 7.02 | 0.16 | n.d. | 0.42 | n.d. | n.d. | 98.8 | Baia Sprie |
| 4748-38 | 33.48 | 58.53 | 4.02 | 0.10 | n.d. | 0.24 | n.d. | n.d. | 96.37 | Baia Sprie |
| SR-4 | 31.6 | 60.16 | 7.73 | 0.10 | 0.29 | 0.12 | n.d. | n.d. | 100 | Şuior |
| SR-6 | 30.4 | 56.52 | 11.5 | 0.86 | 0.66 | 0.06 | n.d. | n.d. | 100 | Şuior |
| SR-8 | 30.9 | 61.95 | 6.92 | 0.20 | n.d. | 0.03 | n.d. | n.d. | 100 | Şuior |
| CS12298_an4 | 34.16 | 60.46 | 5.84 | 0.05 | n.d. | 0.42 | n.d. | n.d. | 100.93 | Cisma |
| CS12298_an9 | 33.79 | 62.74 | 4.11 | 0.03 | n.d. | 0.08 | 0.03 | n.d. | 100.78 | Cisma |
| CS-Cx_an4 | 34.23 | 56.80 | 8.63 | 0.51 | 0.08 | 0.33 | 0.02 | n.d. | 100.6 | Cisma |
| CV-5a_an5 | 34.22 | 59.19 | 6.97 | 0.60 | 0.85 | 0.17 | n.d. | n.d. | 102 | Cisma |
| CV-5_an3 | 33.23 | 62.87 | 1.29 | n.d. | n.d. | 0.58 | n.d. | n.d. | 97.97 | Cisma |
| GB-9_an6 | 32.75 | 63.80 | 1.01 | n.d. | 0.20 | 0.36 | n.d. | n.d. | 98.12 | Cisma |
| BB5B | 32.59 | 63.14 | 1.68 | 0.11 | 0.04 | 0.46 | n.d. | n.d. | 98.02 | Breiner |
| BBXR | 30.43 | 62.78 | 2.73 | 0.44 | 1.02 | 0.55 | n.d. | n.d. | 97.95 | Breiner |
| BBXB | 29.82 | 62.69 | 3.30 | 0.11 | 0.90 | 0.72 | n.d. | n.d. | 97.54 | Breiner |
| BB9B | 30.17 | 61.28 | 4.54 | 0.39 | 1.74 | 0.40 | n.d. | n.d. | 98.52 | Breiner |
| BB806R | 35.14 | 56.68 | 5.10 | 0.09 | n.d. | 0.59 | n.d. | n.d. | 97.6 | Breiner |
| BB877R | 29.77 | 59.80 | 6.41 | 0.20 | 1.98 | 0.35 | n.d. | n.d. | 98.51 | Breiner |

n.d.—not detected

Manganese concentrations are in appreciable quantities, but less than 0.35 wt.%. The largest quantities are in the first generation of sphalerite that range from 0.04 wt.% to 0.35 wt.% (Table 1, Figure 6). The second and third generations of sphalerites have lower contents that range from 0.6 wt.% to 0.10 wt.% and from 0.15 wt.% to 0.16 wt.% respectively. The quantities of manganese correlate with those of iron from the three generations of sphalerite. Cadmium was detected only in sphalerite from the second and third generation in a very low amount that ranges from 0.09 wt.% to 0.13 wt.% (Table 1, Figure 7).

At Turţ Penigher the ore deposit is located in micro-diorite porphyry that intruded the pyroxene lava flow [22]. The vein contains predominantly pyrite, sphalerite, and galena [1]. Sphalerite occurs as homogeneous xenomorphic aggregates associated with pyrite (Figure 3a). Sphalerite is a brown variety with 3.19 wt.%–6.25 wt.% iron and small amounts of manganese, cadmium, and copper (Table 1, Figures 4 and 5).

In the Ilba–Nistru area, the intrusive rocks inside the caldera [32] exercised control over the zonal distribution of the mineralization. The horizontal zoning of the Nistru ore deposit around the quartz-monzodioritic stock (Figure 8) is emphasized:

(1) the centralvein area includes the higher temperature [24] with copper mineralizations + bismuth sulphosalts;

(2) the external area in the periphery of the stock is characterized by the emplacement of the Pb-Zn mineralizations;

(3) gold veins are developed laterally from the margins of the intrusion areas.

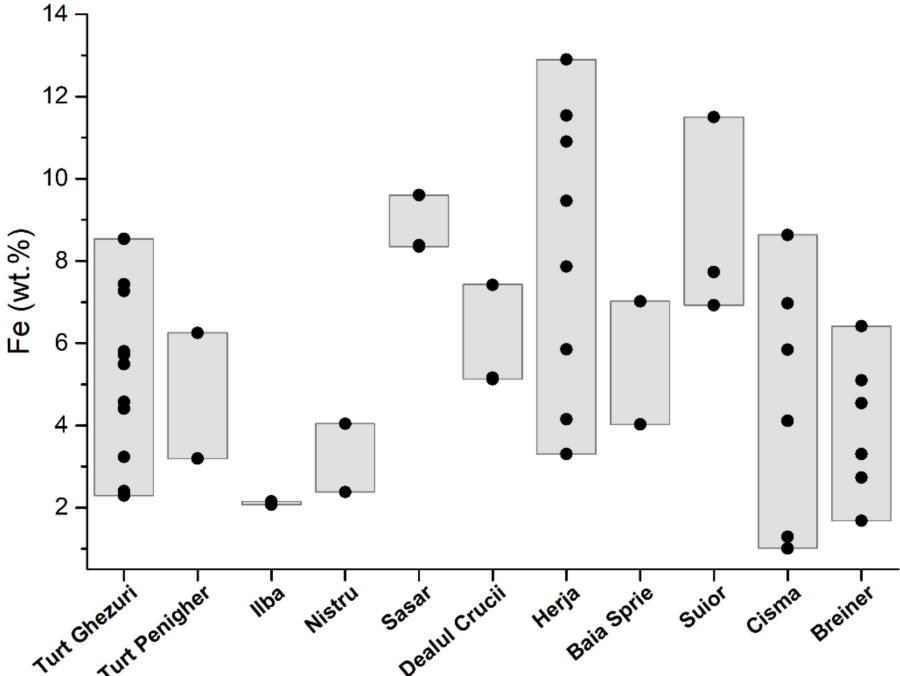

**Figure 5.** Distribution of iron in sphalerite from the Baia Mare and Oaş areas.

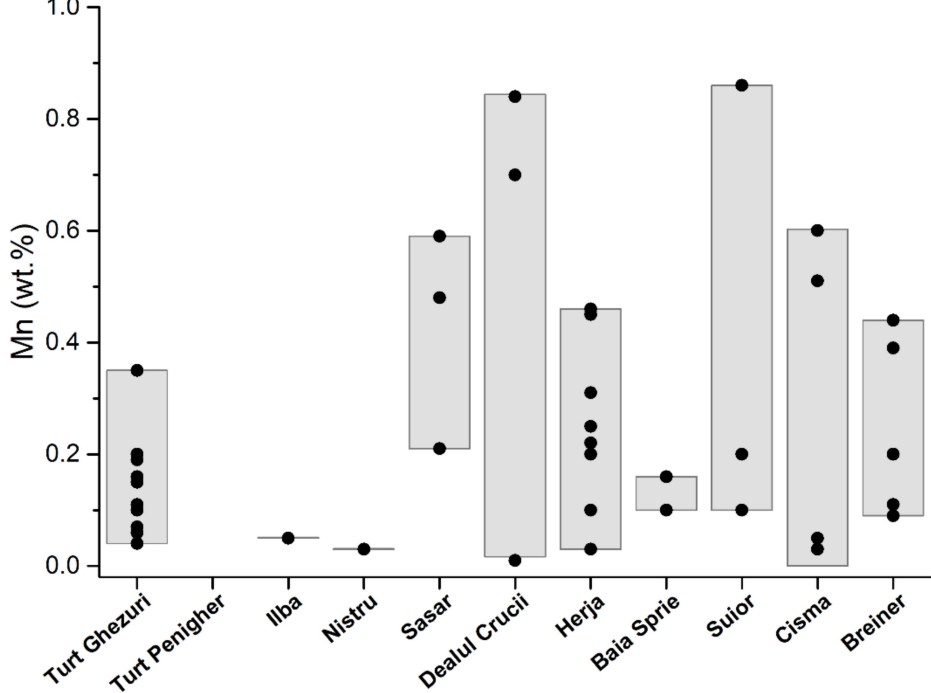

**Figure 6.** Distribution of manganese in sphalerite from the Baia Mare and Oaş areas.

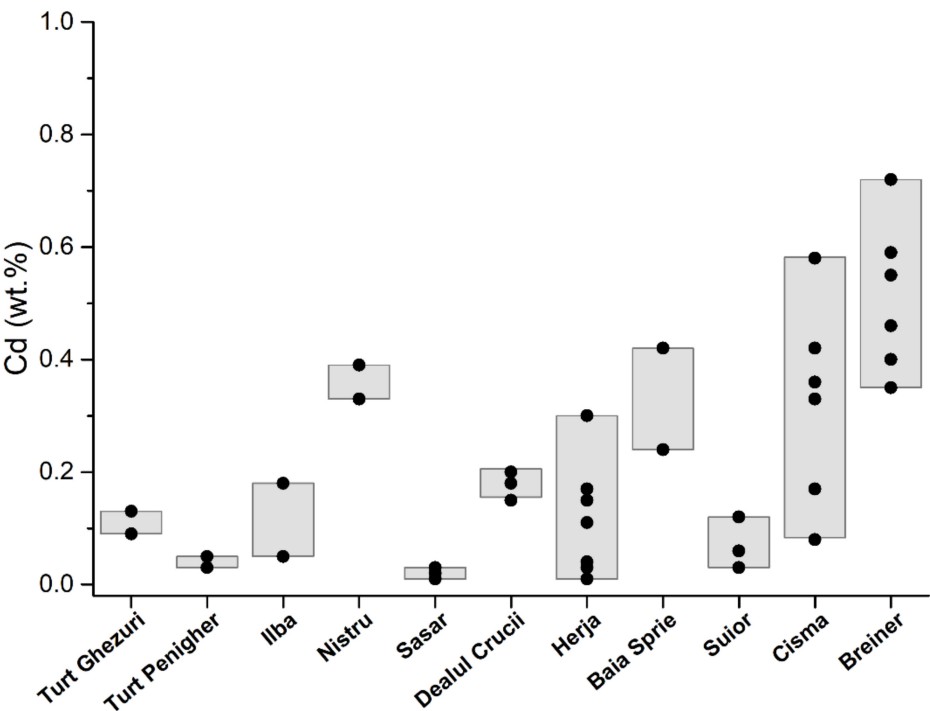

**Figure 7.** Distribution of cadmium in sphalerite from the Baia Mare and Oaş areas.

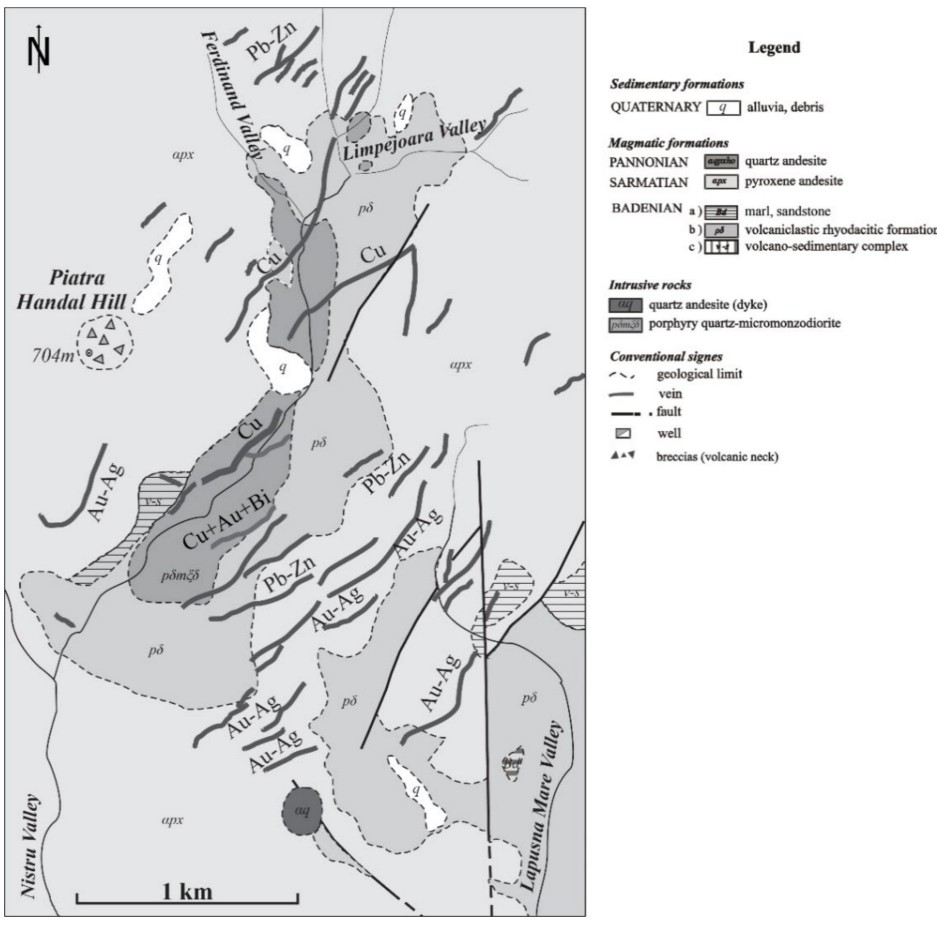

**Figure 8.** Lateral zoning of the mineralization from the Nistru area, with modifications after [25].

The mineralogy of the Ilba–Nistru area is complex, the sulfides prevail in lead–zinc veins. The sphalerite is the main mineral of the base-metal stage and consists of massive ore with brownish color [26,27] associated with galena, pyrite, and chalcopyrite (Figure 2b,c). At the Ilba ore deposit sphalerite appears as xenomorphic granules and intergrows with galena, chalcopyrite, and pyrite (Figure 3b). In the Nistru ore deposit, sphalerite contains exsolutions of chalcopyrite (Figure 3c), which have also been described in [24].

Quantities of iron in base-metal sphalerite range from 2.07 wt.% to 4.04 wt.% (Table 1; Figure 5). These contents are similar to those reported by [24], that range from 2.1 wt.% to 4.8 wt.% of Fe. In these sphalerites the amounts of manganese are low (Table 1; Figure 6). Substantial amounts are those of cadmium that range from 0.05 wt.% to 0.39 wt.% (Table 1, Figure 7), which are similar to those reported by [24], that range from 0.5 wt.% to 1 wt.% of Cd; and 0.1 wt.% to 0.3 wt.% of Mn. After [24] sphalerite of late generation associated with Mn carbonates has elevated Mn content (up to 1.1 wt.%), low Cd (0.2 wt.% to 0.5 wt.%), and variable Fe (1.5 wt.% to 9.2 wt.%).

In the central area from Nistru quartz-monzodioritic, stock sphalerite is associated with pyrrhotite, chalcopyrite, and pyrite. Iron content in sphalerite ranges from 6 wt.% to 16 wt.% in the assemblage with pyrrhotite [24] and from 2.1 wt.% to 10.3 wt.% in the assemblage with pyrite, respectively [24]. It contains small amounts of Cd (0.3 wt.% to 0.6 wt.%) and Mn (0.1 wt.% to 0.3 wt.%) [24].

The most typical stratovolcanic structure is the gold from the mineralization of Săsar (Figure 9). The vein system is made out of veins disposed of by NNW–SSE and NNE–SSW, the second one being more important. The main veins X and XXV are very extended in the vertical direction and pierce the entire stratovolcano structure. In the depth, the intrusive body was intercepted by drillings and veins are located within it.

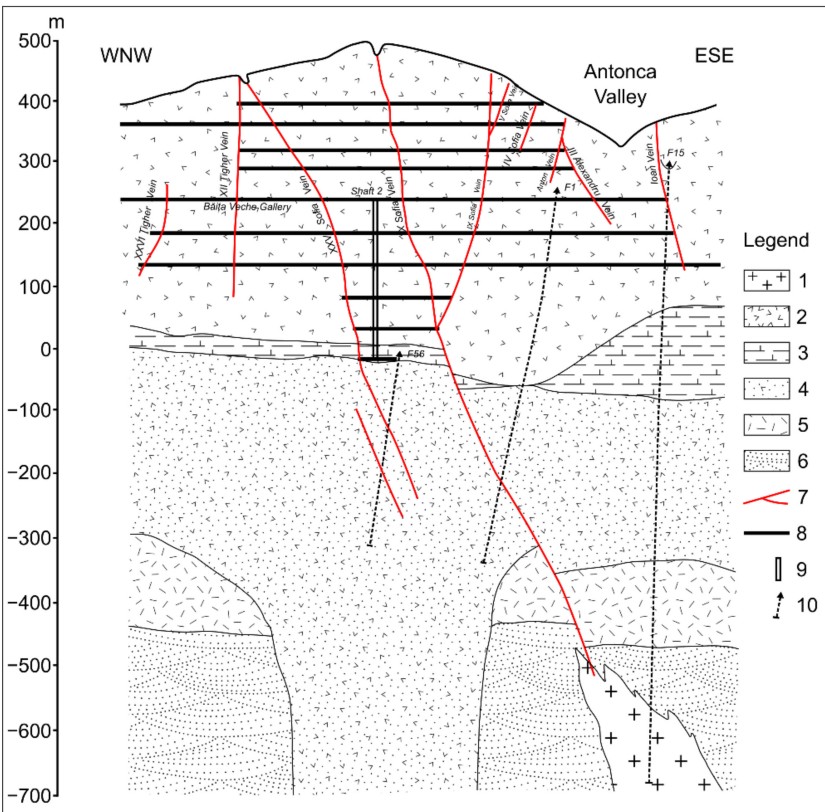

**Figure 9.** Crosssection through Sofia–Propad (Săsar) stratovolcanic structure ([33] with modifications). 1—Porphyry microdiorites; 2—Quartz bearing andesite (Pannonian); 3—Pyroxene andesites (Sarmatiane age); 4—Ryodacite pyroclastic rocks (Lower-Badenian age); 5—Pannonian marls; 6—Badeniane; 7—Eocene sedimentary formations; 7—Veins; 8—Gallery; 9—Shaft; 10—Drillings.

In gold ore deposits from the Săsar area [28], the black-brown sphalerite appears as millimeter and submillimeter granules intergrown with pyrite, chalcopyrite, galena, and quartz [29,34] (Figure 2d). Xenomorphic sphalerite aggregates have numerous chalcopyrite exsolutions and galena–pyrite inclusions (Figure 3d). In the lower areas of the veins located in intrusive bodies, the sulfide content increases. From these areas, sphalerites were sampled for analysis from the veins of Sofia–Săsar, Valea Roşie–Săsar, and Dealul Crucii. Iron content in sphalerite ranges from 5.12 wt.% to 9.60 wt.% Table 1; Figure 5). The manganese contents are very variable and quite large, ranging from 0.21 wt.% to 0.84 wt.% (Figure 6) with one exception which has 0.02 wt.%. Cadmium contents range from 0.01 wt.% to 0.21 wt.% (Figure 7).

Herja ore deposit is the typical base-metal ore deposit in the Baia Mare area, which is characterized by high contents of Zn and Pb. The mineralization consists of a branched system of veins [35]. Sphalerite is frequently associated with pyrite, pyrrhotite, marcasite, and galena (Figure 2e). It frequently appears in the form of black massive masses or bands that alternate with black galena specific to iron-rich varieties. Euhedral crystals up to 5 cm associated with galena and quartz are common. Submillimeter inclusions of pyrrhotite and chalcopyrite are common in black sphalerite. They are characteristic of the first mineralization moments of the base-metal stage formed at high temperatures. Iron content ranges from 3.3 wt.% to 12.9 wt.% (Table 1; Figure 5). Manganese occurs in appreciable quantities and ranges from 0.03 wt.% to 0.46 wt.%, (Table 1; Figure 6). There is a good correlation between iron and manganese. Cadmium contents are low (from 0.01 wt.% to 0.17 wt.%) except for a single sample that has 0.30 wt.% and which correlates with high iron contents (Table 1; Figure 7). The copper detected in sphalerite comes from micrometric chalcopyrite inclusions.

Baia Sprie ore deposit is represented by a vein system associated with intrusive structures. Sphalerite is associated with galena, pyrite, chalcopyrite, and quartz (Figure 3f). Microscopically it contains exsolutions and micrometric inclusions of chalcopyrite at the boundary between the crystals (Figure 2e). Sphalerite is more abundant in the upper part of the deposit where it shows contents of Fe between 4.02 wt.% and 7.02 wt.% [36]. The Cd content varies between 0.24 and 0.42 wt.%, and Mn concentration shows lower values up to 0.16 wt.% (Table 1; Figures 4 and 5). More chemical data on sphalerite composition are available in [36,37].

The sphalerite from the Şuior ore deposit is similar to the one from Herja. The ore deposit is associated with the intrusive body [38]. The mineralization is typically base metal that contains pyrite, sphalerite, galena. Sphalerite is black in color and is intergrown with pyrite, arsenopyrite, galena, and quartz. Small inclusions and exsolutions of pyrrhotite and chalcopyrite are found inside sphalerite (Figure 3g). Contents of Fe range from 6.92 wt.% to 11.5 wt.% and of Mn from 0.10 wt.% to 0.86 wt.% respectively (Table 1; Figures 3 and 4). Cadmium is in a small amount (0.03–0.13 wt.%). The copper detected in sphalerite comes from micronic chalcopyrite inclusions.

The analytical data for sphalerite from Cavnic was focused on M3 (Pb–Zn–Au—sphalerite, galena, native gold, quartz, adularia) and M4 (Pb–Zn–Au–Mn–Sb—rhodonite, rhodochrosite, adularia, galena, sphalerite, tetrahedrite, stibnite, native gold) [39], stages of sphalerite from the Cavnic–Bolduţ zone. Very commonly sphalerite is associated with galena, pyrite, and chalcopyrite (Figure 2h). Microscopically sphalerite is associated with chalcopyrite, pyrite, and galena and contains chalcopyrite exsolutions at the edge of the granules (Figure 3h). M3 stage sphalerite has variable Fe contents (0.8 to 4.8 wt.%), and moderate quantities of Cd and Mn (0.3 to 0.8 wt.% and up to 0.5 wt.% respectively) [24]. Sphalerite of the M4 stage on the contrary is low in Fe (0.1 to 1.6 wt.%) and Cd (0.4 to 0.5 wt.%) and is significantly enriched in Mn (up to 2.9 wt.%) [24].

The Băiuţ area is situated in the easternmost part of the Baia Mare region. We studied the sphalerite from two deposits: Breiner and Cisma. The ore deposits as veins are located in Paleogene sediments intruded by the Pannonian quartz micro-monzodiorite and micro-granodiorite porphyries [40]. The sphalerite appears to be associated and intergrown with chalcopyrite, pyrite, and galena, but it is identified as sphalerites bounded in carbonate [41]. The sphalerite from the Breiner deposit is characterized by low content in Fe that ranges from 1.68 wt.% to 6.41 wt.% (Table 1; Figure 5). The manganese was detected in small amounts (from 0.09 wt.% to 0.44 wt.%) and does not correlate with iron content (Table 1; Figure 6). Cadmium is present in appreciable quantities (from 0.35 wt.% to 0.72 wt.%) (Table 1; Figure 7).

In the Cisma deposit [3], we identified two mineralization stages: the early stage is a copper one; the second is base metal (Figure 2g). Early-stage sphalerites are associated with galena and pyrite and contain very rare chalcopyrite exsolutions (Figure 3f). In the base-metal mineralization in the investigated samples sphalerite is represented by two generations [40]. Sphalerite-1 has Fe contents from 1.52 to 4.62 wt.%, and Mn (0.41 to 0.71 wt.%) and relatively constant Cd contents (approx. 0.2 wt.%). Sphalerite-2 overgrows sphalerite-1, and has similar Fe content (1.3 to 4.5 wt.%), is low in Mn (0.04 to 0.24 wt.%) and Cd (0.21 to 0.45 wt.%) [40]. The iron contents correlate with those of manganese. The determinations made by us indicate the presence of two types of sphalerites: one rich in iron and a second one poor in iron. The first has contents that range from 4.12 wt.% to 8.63 wt.%, and the second sphalerite from 1.02 wt.% to 1.3 wt.% (Table 1; Figure 5).

*5.2. Raman Spectroscopy*

Over the past years, considerable efforts have been expended to determine the concentration of impurities (i.e., Fe, Mn, Cd, Co, Cr, etc.) by means of Raman spectroscopy, which commonly substitute for zinc in sphalerite lattice sites [37,42–45]. The assignment of the Raman modes is based on the previous studies [37,44,45]. Briefly, the first-order Raman spectrum is split into two fundamental modes: (i) LO mode (longitudinal optical) at ~350 cm$^{-1}$, and (ii) TO mode (transverse optical) at ~272 cm$^{-1}$. Lower wavenumbers (below 220 cm$^{-1}$) consist of acoustic modes, while the combination modes are found at wavenumbers higher than 390 cm$^{-1}$.

The Raman spectra of the sphalerite samples from the Baia Mare area (Figure 10) show typical Raman modes with the main peaks at 300, 331, and 350 cm$^{-1}$. Since the Fe content in sphalerites ranges from 1.01 wt.% to 12.9 wt.%, the Raman bands exhibit variable intensities. In samples where the Fe concentration is higher, the intensities of 300 and 331 cm$^{-1}$ peaks are increasing while the 350 cm$^{-1}$ mode is diminishing.

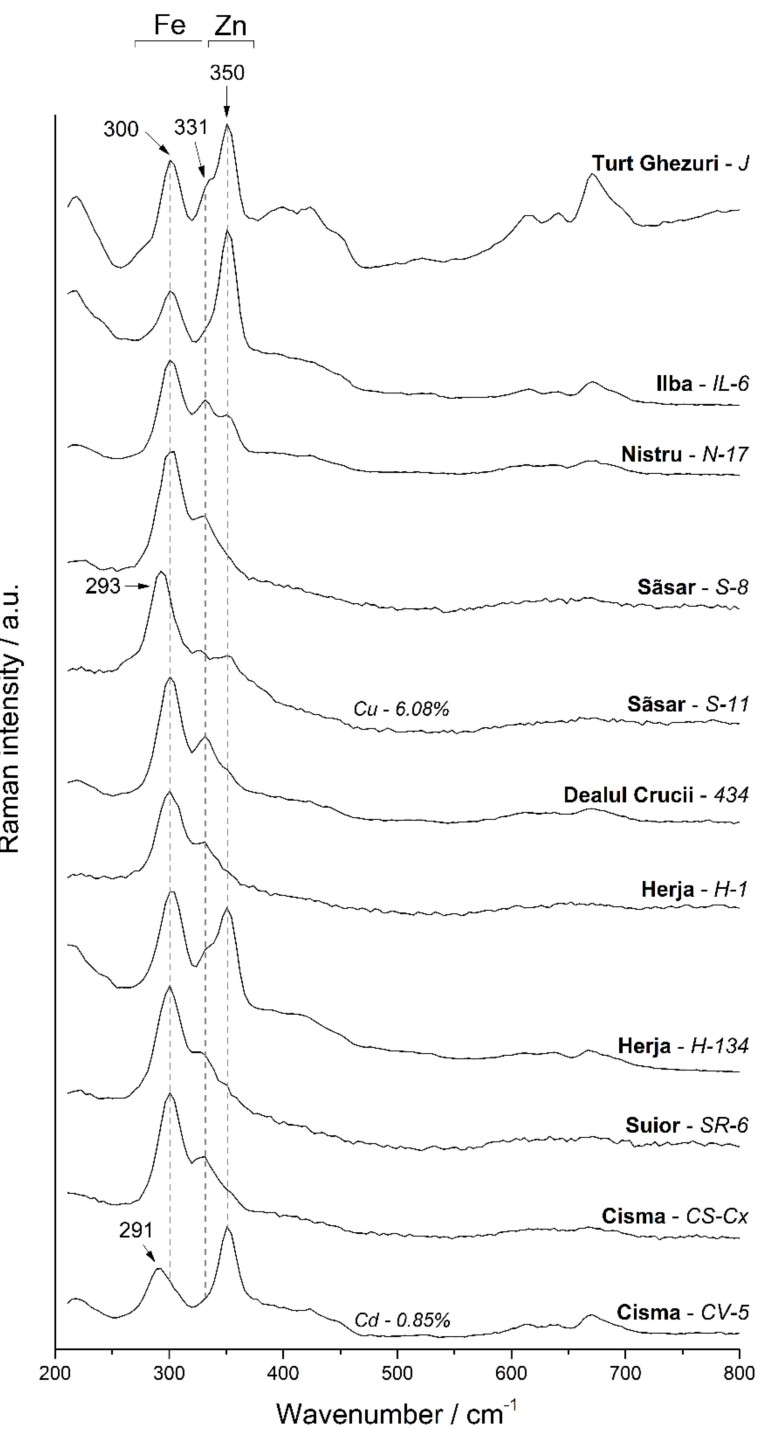

**Figure 10.** Selective Raman spectra of the studied sphalerite samples.

## 6. Discussion

The presence of great quantities of Fe has been observed in the structure of sphalerite in the mineralizations from the Baia Mare mining district (Table 1). The sphalerites from Baia Sprie and some from Cavnic, Iba, Turț Penigher, and partial at Băiuț Cisma and Breiner have relatively low contents. In contrast, there are great contents of Fe at Herja and partial at Ghezuri and Nistru (copper stage) where the sphalerite is associated with the pyrrhotite. Surprisingly, the sphalerite from the gold mineralizations from the Săsar area has high quantities of Fe close to the ones from Herja to which they are alike from this point of view.

The maximum iron content in sphalerites in the Baia Mare area is 12.9 wt.%. Ref. [10] showed iron concentrations reaching up to over 15% by weight. In epithermal, low-sulfidation mineralizations in the Apuseni Mountains (Golden Quadrilater Romania) with mineralizations of Au veins and magmatic breccia and significant base-metal ore, sphalerites contain iron between 0.06 and 10.44 wt.% [10]. These contents are much lower than those in the metallogenetic districts of Baia Mare and Oaş (Table 1). Similar contents (1.31–8.48 wt.%) were reported for the Čumavići polymetallic ore deposit (Bosnia) by [46].

Substitution of zinc with minor elements in the structure of sphalerite depends on the similarity of the covalent rays of those elements and their affinity for tetrahedral coordination. The correlation between iron and zinc from sphalerite from the metallogenetic district of Baia Mare and Oaş is strongly negative (Figure 11). The negative correlation shows that iron is the main element that replaces zinc in the sphalerite structure. This substitution is possible because the cation radii of zinc and iron are similar ($Fe^{2+}$ = 1.25 Å; $Zn^{2+}$ = 1.31 Å) which favors the substitution of zinc. The interatomic distance at Fe-S is greater than at Zn-S [4]. Isomorphic substitution of zinc with iron causes an increase in the reticular parameter *a* of sphalerites [47] with increasing iron content. Kullerud (1953) [4] and Barton and Kullerud (1958) [48] indicated that the ability of FeS to replace ZnS depends on the increase in temperature.

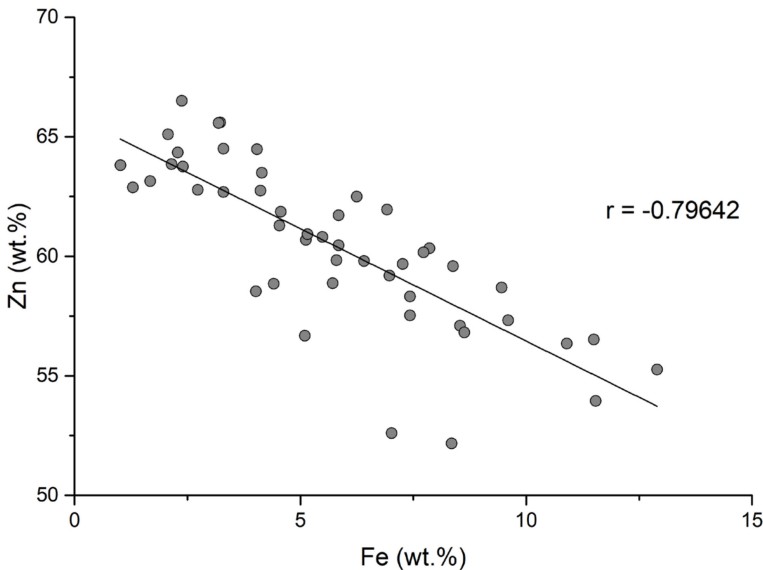

**Figure 11.** Correlation between Zn and Fe concentrations in sphalerite from the Baia Mare and Oaș areas.

Subsequently, the authors of [5,49] reported a nonlinear behavior depending on the Fe content over ~10 mol%, which was correlated with variable sulfur fugacities. Higher sulfur leakages were correlated with elevated $Fe^{3+}$ concentrations [49].

The highest manganese content of sphalerites in the Baia Mare and Oaş area is 0.84 wt.% (in the Dealul Crucii epithermal gold deposit) (Table 1). The correlation between Zn and Mn (Figure 12) is negative indicating that the incorporation of manganese takes place by simple cation exchange ($Zn^{2+} \leftrightarrow Mn^{2+}$). The incorporation of MnS into the sphalerite structure is in the form of a solid solution. Alabandite MnS is not isostructural with sphalerite and the manganese contents entering the sphalerite structure cannot be very high. According to the data presented by [50], the MnS contents of the sphalerite structure cannot exceed 7 mol%. There is a good correlation between iron and manganese (Figure 13), indicating that the substitution of $Zn^{2+} \leftrightarrow Fe^{2+}$ is accompanied by $Zn^{2+} \leftrightarrow Mn^{2+}$. These double substitutions are possible because the cation radii of zinc, iron, and manganese are similar ($Zn^{2+}$ = 1.31 Å; $Fe^{2+}$ = 1.25 Å; $Mn^{2+}$ = 1.39 Å). Cook et al. (2009) [10]

reported a content of 6.7 wt.% Mn in sphalerite in epithermal gold deposit from Roşia Montana (Romania). These contents exceed the maximum value indicated by [50].

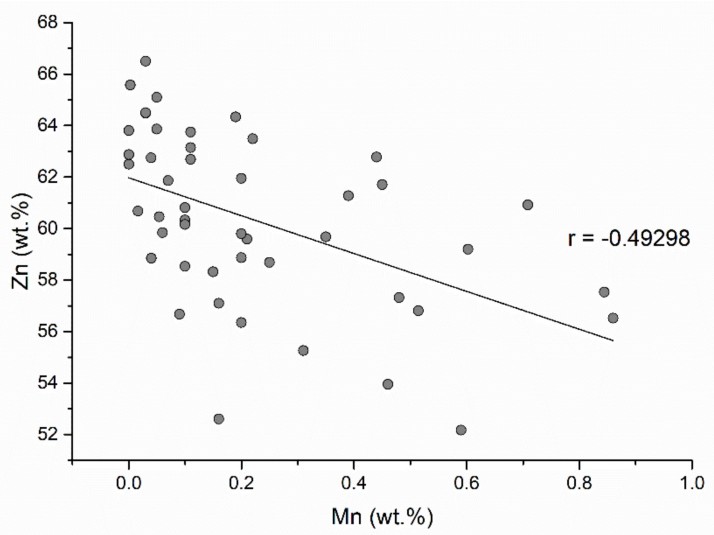

**Figure 12.** Correlation between Zn and Mn concentrations in sphalerite.

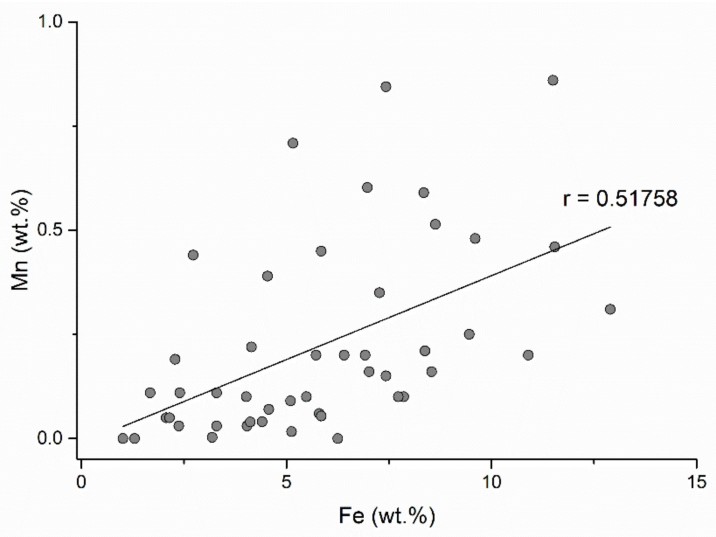

**Figure 13.** Correlation between Fe and Mn concentrations in sphalerite.

Cadmium frequently occurs as a solid solution in sphalerite. The cadmium content is quite uniform in the Baia Mare and Oaş areas with contents ranging from 0.01–0.72 wt.%. The correlation between Cd and Fe and Cd and Mn (Figures 14 and 15) is negative and indicates that cadmium replaces zinc in sphalerite with low iron and manganese contents. Cadmium replaces zinc due to the similarity of their covalent rays ($Zn^{2+}$ = 1.31 Å; $Cd^{2+}$ = 1.48 Å) and the sulfur affinity of cadmium which is similar to that of zinc. The correlation between cadmium and manganese (Figure 15) is insignificant because these elements independently replace iron in sphalerite. Iron-rich sphalerites from Herja, Şuior, and Săsar contain small amounts of cadmium. Another peculiarity is sphalerite from gold mineralization from Dealul Crucii, which contains quite a lot of iron and also contains appreciable amounts of cadmium. Additionally, sphalerites from the eastern part of the Baia Mare area (base-metal and copper mineralization from Cisma and Breiner) incorporate the highest amounts of cadmium.

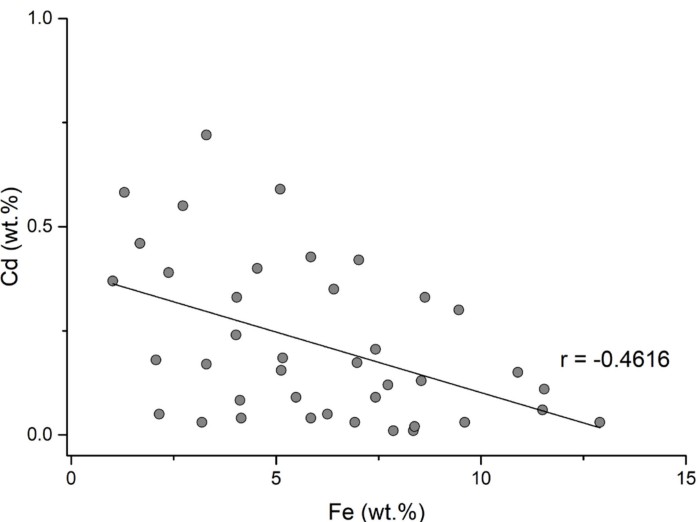

**Figure 14.** Cd and Fe correlation in the studied sphalerites.

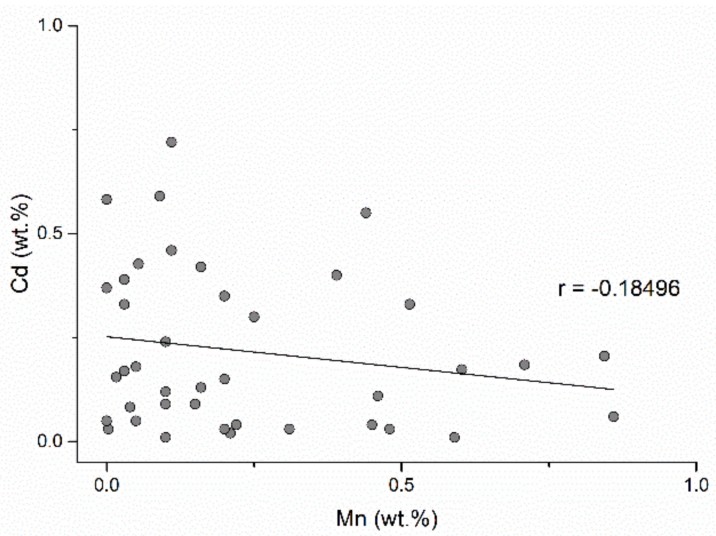

**Figure 15.** Cd and Mn correlation in the studied sphalerites.

The copper contents in sphalerite are low. It can replace iron in the sphalerite structure or it can be present in the form of micron exsolutions barely visible under a microscope. Higher contents were determined at the deposits in the Baia Mare area at Băiuţ, Cisma, and sometimes at Dealul Crucii and Herja. These contents resulted from chalcopyrite exsolutions of sphalerite, which are also visible under the microscope [40].

Low concentrations of silver and indium were detected in samples from the goldfield Dealul Crucii, being an important feature for these sphalerites. This deposit is characterized by the presence of sulphosalts [34] and high formation temperatures.

Raman spectroscopy has proved to be very sensitive in highlighting the structural and chemical changes in sphalerites. Several studies [37,42,43] have shown a direct correlation between relative Raman intensities and cation concentration, which can be very effective to quantify the levels of cation substitution in sphalerites. For the estimation of the Fe content in natural sphalerites, the intensities of all three peaks (i.e., 300, 331, and 350 cm$^{-1}$) are correlated with the iron content [37]. The Raman spectra of Mn-bearing sphalerites established by [43] on synthetic sphalerites (0.03–0.5 wt.% Mn), also show a correlation between peak intensities and Mn concentration. Due to the influence of Cd substitution on Raman vibrational characteristics in synthetic sphalerite, the authors of [42] suggest a new Raman active mode (*X* mode) occurring at 295 cm$^{-1}$ (Cd–S bond vibrations).

Previous studies showed that there is a linear correlation between these intensities and therefore they can be used for semi-quantitative determinations [37]. For the sphalerite samples with Mn content in the crystal lattice, the Raman peaks become slightly broader with increasing Mn concentration, which is in agreement with the observations in [43]. Two Raman spectra revealed a slight shifting of the 300 cm$^{-1}$ modes to lower wavenumbers (Figure 10, CV-5 and S-11 samples). This might be due to the presence of additional cations in the chemical composition, since these samples have higher contents of Cu (6.08 wt.%)—S-11 and Cd (0.85 wt.%)—CV-5, respectively. The influence of Cd on the Raman spectra of sphalerite was reported by [42,44] showing that the peak assigned to Cd–S bonds is found at 285–295 cm$^{-1}$. Nevertheless, in natural sphalerites, coupled substitutions are present due to the complexity of natural hydrothermal systems. Therefore, this mode could be affected not only by Cd but also by other several cations present in the crystal structure producing this shift to lower wavenumbers.

The Fe content of sphalerites is known as an important indicator of the physico-chemical conditions of deposit formation [30] because it is a function of temperature, pressure, and sulfur fugacity.

The sphalerite composition has been proposed to determine the temperature of mineralization formation. The studies made by [4,48] show that the $X_{FeS}$ content in the structure of sphalerite depends on temperature. The geothermometer based on the sphalerite composition depends on the ability of FeS to substitute ZnS depending on the temperature in the FeS-ZnS system. Refs. [4,48] propose the use of the solvus curve in mixed crystals (Zn, Fe) S and FeS to determine the temperature of the formation of the sphalerite-pyrrhotite association. Subsequent studies [5,6], had disapproved previous research, sphalerite in equilibrium with pyrite and pyrrhotite contains increasing amounts of FeS, as the temperature drops below 742 °C to 550 °C, where it reaches 20.8 ± 0.5% moles of $X_{FeS}$, which remains constant up to 303 °C. These authors noted the importance of temperature and S fugacity on the extent of solubility. Chernyshev and Anfilogov (1968) [51] and Einaudi (1968) [52] suggest that at 300 °C the $X_{FeS}$ content in sphalerites would be 26 ± 3 wt.%, and 32.2 ± 1.2 wt.%, respectively. The data presented in [5,6] for the temperature range from 742 °C to 550 °C remain valid. According to the authors of [8] the use of the geothermometer based on the composition of sphalerites can be used to determine the temperature of formation of sphalerites associated with pyrite from metamorphic ore deposits, using the 7.5 kb isobar, being inapplicable for sphalerites associated with pyrite and pyrrhotite whose $X_{FeS}$ are high.

Within the ore deposits from the Baia Mare area, there is no correlation between the $X_{FeS}$ contents in sphalerites and the values of fluid inclusion homogenization temperatures (Table 2) in ore deposits from the Baia Mare area. The formation temperatures determined based on fluid inclusions studies are similar for the Herja–Baia Sprie, Săsar–Ilba, or Turţ Ghezuri–Cavnic ore deposits, yet the contents of $X_{Fes}$ in sphalerite are quite different. A similar situation is observed between Turţ Ghezuri and Turţ Penigher ore deposits which are spatially close to each other. All these inconsistencies show that the $X_{FeS}$ content of sphalerites in epithermal mineralization does not depend on the formation temperatures and the composition of sphalerites can only be used as a geothermometer with great caution.

**Table 2.** Chemical composition of sphalerite and homogenization temperature.

| No. | Ore Deposits | Fe in Sphalerite wt.% | | Temperature | | References |
| --- | --- | --- | --- | --- | --- | --- |
| | | Max | Min | Max °C | Min °C | |
| 1 | Ghezuri-Turţ | 8.54 | 2.40 | 300 | 260 | [53] |
| 2 | Penigher-Turţ | 6.25 | 3.19 | 287 | 260 | [53] |
| 3 | Ilba | 2.15 | 2.07 | 360 | 200 | [54] |
| 4 | Nistru [1] | 16.00 | 2.01 | 300 | 200 | [24] |
| 5 | Sasar | 9.60 | 5.12 | 350 | 250 | [33] |
| 6 | Herja | 12.90 | 3.30 | 300 | 250 | [23] |
| | | | | 350 | 200 | [35] |
| 7 | Baia Sprie [2] | 7.00 | 4.00 | 300 | 240 | [23] |
| | | | | 345 | 190 | [55] |
| 8 | Şuior | 4.80 | 0.80 | 280 | 230 | [23] |
| 9 | Cavinic [3] | 4.80 | 0.10 | 298 | 250 | [56] |
| 10 | Băiuţ | 6.41 | 1.68 | 360 | 230 | [57] |
| | | | | 357 | 228 | [58] |
| 11 | Cisma [1] | 8.63 | 1.30 | 344 | 143 | [40] |

[1] [24]; [2] [37]; [3] [40].

In the field of temperatures of high metallogenetic interest 300–600 °C, the $X_{FeS}$ of sphalerite associated with pyrrhotite and pyrite is a function of pressure [7–9,59]. Thus, it is possible to use sphalerite as a geobarometer for the entire range of FeS content variation. Complete data on the pressure dependence of $X_{FeS}$ in sphalerites result from the research in [9]. According to the authors of [59], the curves of the isobaric composition for sphalerite associated with pyrite and hexagonal pyrrhotite show a progressive decrease in iron content with increasing pressure and decreasing temperature and it can be used as a geothermometer.

In the Baia Mare area, sphalerite is associated with hexagonal pyrrhotite only in Herja [35]. Sphalerite from Herja, Nistru, Săsar, and Cisma has high iron content (Table 1) which corresponds to those reported in [59] with medium pressure and high formation temperatures. The high temperature of formation in these deposits is correlated with high amounts of $X_{FeS}$ (Table 2). Instead, according to the experiments in [59], the pressure should be over 2.5 kb. These pressures are far too high compared to those determined by the authors of [40,60] of maximum 1 kb suggesting a formation depth of about 500 to 800 m for a hydrostatic regime and cannot be accepted for hydrothermal deposits in the Baia Mare area. Other authors [49,61] also emphasized the need to pay close attention to the use of this geobarometer due to compositional and textural readjustments in hydrothermal sulfides.

The iron content in sphalerites also depends on the $fS_2$ [49]. Higher concentrations of Fe are observed in samples from Herja and partially from Nistru, Săsar, Cisma, and Ghezuri, caused by the higher fugacity of $fS_2$ [24]. In the epithermal deposits, the superficial pressure seems to be unimportant and the composition of the sphalerite can be discussed in terms of the $fS_2$–T relation [40]. Several authors have suggested that the boiling of hydrothermal fluids in the Baia Mare area contributes to the forming of deposits [35,36,39]. Iron content decreases with $fS_2$ and temperature decreases. The low contents of Turţ, Ilba, Nistru, Baia Sprie, Cavnic, and Băiuţ suggest low forming temperatures of mineralization and the reduction of $fS_2$. The high iron content of the sphalerites in the gold ore deposits in the Săsar area could be formed at high $fS_2$ which explains the lack of tellurium and the presence of native gold. If the temperature remains relatively constant, the boiling of the fluid will lead to loss of sulfur gas, meaning a decrease in $fS_2$. If the temperature drops from 350 to 150–200 °C, as estimated for all deposits in the Baia Mare area, and if $fS_2$ remains constant, the $X_{FeS}$ of the sphalerite will decrease slightly. In many deposits in the Baia Mare area, $fS_2$ can also decrease during cooling due to sulfur deposition or due to dilution by meteoric waters.

In some deposits such as Herja, Suior, and Sasar, sphalerite deposition began in the pyrrhotite stability field and stopped in the that of pyrite–chalcopyrite [24]. However, the content and temperature decrease from >300 to 200 °C (Table 2), suggesting forming conditions near the pyrite/pyrrhotite buffer. A similar trend is observed at the deposits from Cisma, Nistru, Ghezuri. Sphalerite from the Cavnic, Baia Sprie, Ilba, and sometimes Nistru deposits have the lowest $X_{FeS}$, which may be the result of a small decrease in fugacity ($fS_2$).

For the Baia Mare area, we can appreciate that the sphalerite was formed during the decrease of $X_{FeS}$ from the beginning of the deposition process until the later generations, accompanied by the decrease of the temperature and a slight decrease of the sulfur fugacity.

## 7. Conclusions

This study presents a detailed approach of sphalerite from base-metal epithermal mineralizations from the Baia Mare and Oaş areas. In the Baia Mare and Oaş areas, there are gold–silver and base-metal epithermal ore deposits with significant amounts of sphalerite. In these mineralizations, sphalerite paragenesis is complex, forms a massive texture, and has been largely deposited by boiling.

Large amounts of Fe were observed in sphalerite in all mineralizations in the mining district of Baia Mare and Oaş. The sphalerites from Baia Sprie, Cavnic, Iba, Turţ Penigher, and partially at Băiuţ, Cisma, and Breiner have a low content of Fe. High iron contents are at Herja and partially in Ghezuri and Nistru (copper stage) where sphalerite is associated with pyrrhotite. Sphalerite from gold mineralization in the Săsar area has large amounts of Fe close to that of Herja. High iron contents are in the deposits associated with subvolcanic intrusions.

The manganese content of the sphalerites in the Baia Mare and Oaş area is of a maximum of 0.84 wt.% (in the Dealul Crucii gold epithermal deposit). The cadmium content is uniform in the Baia Mare and Oaş area, between 0.01 and 0.72 wt.%. The correlation between Cd–Mn and Cd–Fe is negative and indicates that cadmium replaces zinc in sphalerites with a low iron and manganese content. Small contents of silver and indium were detected in the Dealul Crucii goldfield.

Raman spectroscopy has proved to be very sensitive in highlighting the structural and chemical changes in sphalerites. Besides the variation in intensities of the main peaks (300, 331, and 350 cm$^{-1}$) due to the Fe–Zn substitutions, it was observed that the 300 cm$^{-1}$ Raman line is shifted to lower wavenumbers or it appears as a new mode at 291–293 cm$^{-1}$. In natural sphalerites, coupled substitutions are present due to the complexity of natural hydrothermal systems. Therefore, this mode could be affected not only by Cd but also by other several cations present in the crystal structure.

The Fe content of sphalerites is known as an important indicator of the physico-chemical conditions of deposit formation because it is a function of the temperature, pressure, and sulfur fugacity. The composition of the sphalerite was proposed as a criterion to determine the temperature and pressure of the formation of deposits. The $X_{FeS}$ content of sphalerites in epithermal deposits does not depend on the formation temperatures. $X_{FeS}$ from sphalerites associated with pyrrhotite and pyrite is a function of pressure. Thus, it is possible to use sphalerite as a geobarometer for the full range of variations in FeS content. In the Baia Mare area, sphalerite is associated with hexagonal pyrrhotite only at Herja and sporadically at Nistru. Sphalerite from Herja, Nistru, Săsar, Cisma has high Fe content (Table 1) which corresponds with medium pressure and high formation temperatures.

The iron content of sphalerites also depends on the sulfur fugacity. The high iron content in sphalerite is present only at Herja and partially at Nistru, Sasar, Cisma, and Ghezuri, caused by the higher fugacity of $fS_2$. The low content of iron in sphalerites from Turţ, Ilba, Nistru, Baia Sprie, Cavnic, and Băiuţ suggests low temperatures of mineralization formation and the reduction of fugacity $fS_2$.

For the Baia Mare area, we can appreciate that the sphalerite was formed during the decrease of $X_{FeS}$ from the beginning of the deposition process until the later generations, accompanied by the decrease of the temperature and a slight decrease of the sulfur fugacity.

Sphalerite composition from hydrothermal mineralizations is extremely important in ore processing and recovery of minor elements.

**Author Contributions:** Conceptualization, G.D.; methodology, G.D., A.B., A.I.A., F.D., and A.E.M.; software, A.B. and A.I.A.; validation, G.D., A.B., and A.I.A.; formal analysis, G.D., A.B., A.I.A., F.D., and A.E.M.; investigation, G.D., A.B., and A.I.A.; resources, G.D., A.B., and A.I.A.; data curation, G.D.; writing—original draft preparation, G.D., A.B., and A.I.A.; writing—review and editing, G.D., A.B., A.I.A., F.D., and A.E.M.; visualization, A.B. and A.I.A.; supervision, G.D.; project administration, G.D. All authors have read and agreed to the published version of the manuscript.

**Funding:** This research received no external funding.

**Acknowledgments:** The authors are grateful to the staff of the State Geological Institute of Dionýz Štúr (Bratislava, Slovakia) for facilitating access to EPMA.

**Conflicts of Interest:** The authors declare no conflict of interest.

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
