# Peer review of "Hydrothermal Sphalerites from Ore Deposits of Baia Mare Area"

_minerals, doi:10.3390/min11121323_

Round 1

Reviewer 1 Report

Review of manuscript minerals-1451382

Damian G., Buzatu A., Apopei A.-I., Damian F., Maftei A.-E.

Hydrothermal Sphalerites from ore deposits of Baia Mare area

Special Issue: Sphalerite Composition and Formation Conditions in Epithermal Deposits

The article discusses the content of trace impurities in the sphalerite of the Baia Mare and Oas ore area. EPMA and Raman spectroscopy were used as research methods. The article needs some rework.

Some remarks

Sections “Geological and petrogenesis data” and “Metallogeny of the Baia Mare and OaÅŸ area”

The sections need to be simplified and maybe merged. Now it is difficult to read it without referring to other sources. The attached diagram Fig. 1 does not help perception.

Section “Mineral paragenesis and geochemistry of Sphalerite”

It seems to me that the sections need to be divided - 1) Mineral parageneses and 2) Geochemistry of sphalerite.

When characterizing mineral parageneses, it is necessary to illustrate them with photographs and micrographs of ores. It might be a good idea to organize the data on the studied sphalerite samples in tabular form.

The geochemistry of sphalerite was examined using an unrepresentative sample. As I understand it, this is about 50 analyzes for 11 ore objects. In this case, it is better to represent the graphs of the distribution of impurities in sphalerite as a set of points, and not as a box-plot.

Section “Raman spectroscopy”

Results and discussion mixed. In the discussion, the results of spectroscopy were not reflected in any way. Therefore, it is necessary to separate the results and discussion in this part.

Discussion

The statistical calculations for impurities in sphalerite do not seem convincing due to insufficient data.

The substitution of iron for zinc is a fact from mineralogy textbooks. It hardly needs to be discussed. The same, however, applies to other elements considered.

The correlation between cadmium and manganese in sphalerite (Fig. 13) is most likely insignificant.

Section “Implications in mineral processing”.

Perhaps everything is correct, but the parameters of the unit cell of sphalerite were not considered and the surface properties were not studied. So why discuss this in an article?

The arrangement of the pictures is unsuccessful. It is better to collect sections of deposits in one figure, diagrams in another.

The sum of analyzes is missing in table 1.

I hope that these considerations and comments will prove useful

Author Response

Response to Reviewer 1 Comments

Thank you for your comments and suggestions that have helped us improve our article.

The article discusses the content of trace impurities in the sphalerite of the Baia Mare and Oas ore area. EPMA and Raman spectroscopy were used as research methods. The article needs some rework.

Some remarks

Point 1. Sections “Geological and petrogenesis data” and “Metallogeny of the Baia Mare and OaÅŸ area”

The sections need to be simplified and maybe merged. Now it is difficult to read it without referring to other sources. The attached diagram Fig. 1 does not help perception.

Response 1: Sections “Geological and petrogenesis data” and “Metallogeny of the Baia Mare and OaÅŸ area” were simplified taking into account the suggestions of the other reviewers and editor.

Point 2. Section “Mineral paragenesis and geochemistry of Sphalerite”

It seems to me that the sections need to be divided - 1) Mineral parageneses and 2) Geochemistry of sphalerite. When characterizing mineral parageneses, it is necessary to illustrate them with photographs and micrographs of ores. It might be a good idea to organize the data on the studied sphalerite samples in tabular form.

Response 2: Mineral parageneses, was completed with illustrated them with macrophotographs and micrographs of ores and with the text description of paragenesis.

Point 3. The geochemistry of sphalerite was examined using an unrepresentative sample. As I understand it, this is about 50 analyzes for 11 ore objects. In this case, it is better to represent the graphs of the distribution of impurities in sphalerite as a set of points, and not as a box-plot.

Response 3:For the studied area we consider that the analyzes made by the author are representative.

Point 4. Section “Raman spectroscopy”

Results and discussion mixed. In the discussion, the results of spectroscopy were not reflected in any way. Therefore, it is necessary to separate the results and discussion in this part.

Response 4: The synthesis of Raman Spectroscopy results was introduced in the discussions. We have separated the results and discussion in two separate chapters.

Point 5. Discussion

The statistical calculations for impurities in sphalerite do not seem convincing due to insufficient data. The substitution of iron for zinc is a fact from mineralogy textbooks. It hardly needs to be discussed. The same, however, applies to other elements considered.

The correlation between cadmium and manganese in sphalerite (Fig. 13) is most likely insignificant.

Response 5: Statistical calculations have been revised. Correlation between cadmium and manganese in sphalerite is not significant, we have specified in the text.

Point 6. Section “Implications in mineral processing”.

Perhaps everything is correct, but the parameters of the unit cell of sphalerite were not considered and the surface properties were not studied. So why discuss this in an article?

Response 6: Implications in processing sphalerite has been eliminated

Point 7.The sum of analyzes is missing in table 1.

Response 7: The table was completed with the sum of the analyzes.

Point 8. I hope that these considerations and comments will prove useful

Response 8: We tell you that your considerations and comments were useful in improving our article.

Reviewer 2 Report

Dear author(s),

I have completed reveiwing your MS. I made some corrections, suggestions and raised some questions, all marked attached MS.

Implementation of those will enhance overall quality and scientific soundness of the article. I would like see some photomicrographs of the mineral paragenesis, but none included. 

regards 

Author Response

Response to Reviewer 2 Comments

The corrections made in the article in PDF format have been revised as follows:

p.1, line 30; We changed the epithermal deposits with hydrothermal ore deposits.

  1. 1, line 40; We cited article 9
  2. 5 line 199 We cut this with both
  3. 6, line 2014-2015, We cut the last line
  4. 6, line 2018, cubanite occurs as an exsolution in chalcopyrite
  5. 9, line 250, We cited: Plotinskaya2014

p.13, line 384, we wrote: relatively low contents

p.14, line 393, I cited the author [10], It was not our analysis

p.15, line 432, We cut the: covalent rays and wrote: cation radii

p.16, line 473, We cut by

p.17, line 492-495, We cut these lines

9.17, line 513, We cut phase and wrote field, and cut one and wrote that

Reviewer 3 Report

Dear Authors: The manuscript titled “Hydrothermal Sphalerites from ore deposits of Baia Mare area” by Gheorghe Damian et al., aimed at analyze the contents of iron and other minor elements in sphalerite to show the conditions of formation. The work presents a large amount of new data apparently all of good quality, I recommend the publication of this text by Minerals after  revisions.

Here are my major comments:

  1. English expression is the most prominent problem, including grammar and professional terms (For example: Lines135-136: Reviewing these data…the second one in the eastern area: Herja-BăiuÅ£; Line 193:“most frequent”ï¼› K-Ar dating”and“Ar-Ar dating”ï¼›Lines 196-197: In some mineralizations these parageneses are much more complex)ï¼›
  2. The main research object of this article is sphalerite in the hydrothermal deposit in Baia mare area. The mineralogical characteristics (classification, micrograph, etc.) of sphalerite should be described in detail, but the author did not provide any micrograph of sphalerite and a detailed description of classification;
  3. The marking method of references in this paper should meet the requirements of journal;
  4. Part 3: The structure is chaotic, with repeated parts (lines 99-104 and lines 124-135 are repeated), and lack of introduction to mineralogy;
  5. Lines 201-202: “This mineral can offer information about the change in the chemical composition of the mineralizing fluids.”, Please provide evidence or corresponding references.
  6. Lines 214-215:“At least three generations of sphalerites can be separated, which correspond to the associations listed above.”, this part only divides the three generations of sphalerite formation according to the vertical zoning of mineral assemblage, which should provide more evidence;
  7. In Table 1, sphalerite should be classified into different types in addition to ore deposits, because many of the analysis results are described according to the type of sphalerite.

Author Response

Response to Reviewer 3 Comments

Point 1. English expression is the most prominent problem, including grammar and professional terms (For example: Lines135-136: Reviewing these data…the second one in the eastern area: Herja-BăiuÅ£; Line 193:“most frequent”ï¼› K-Ar dating”and“Ar-Ar dating”ï¼›Lines 196-197: In some mineralizations these parageneses are much more complex)ï¼›

Response 1: The English expression including grammatical and professional terms has been revised. Rows 135-136: 193: and 196-197; they have been revised to be much clearer.

Point 2. The main research object of this article is sphalerite in the hydrothermal deposit in Baia mare area. The mineralogical characteristics (classification, micrograph, etc.) of sphalerite should be described in detail, but the author did not provide any micrograph of sphalerite and a detailed description of classification

Response 2: Microphotographs were introduced into the text which were described in detail in the text.

Point 3. The marking method of references in this paper should meet the requirements of journal

 Response 3: The references were verified to meet the requirements of the journal.

Point 4. Part 3: The structure is chaotic, with repeated parts (lines 99-104 and lines 124-135 are repeated), and lack of introduction to mineralogy;

 Response 4: The chaotic structure with repeated parts in rows 99-104 and 124-135 has been corrected and new data on mineralogy have been introduced.

Point 5. Lines 201-202: “This mineral can offer information about the change in the chemical composition of the mineralizing fluids.”, Please provide evidence or corresponding references.

Response 5: A citation was introduced to these lines.

Point 6. Lines 214-215:“At least three generations of sphalerites can be separated, which correspond to the associations listed above.”, this part only divides the three generations of sphalerite formation according to the vertical zoning of mineral assemblage, which should provide more evidence;

Response 6. The text from the 2014-2015 lines has been revised.

Point 7. In Table 1, sphalerite should be classified into different types in addition to ore deposits, because many of the analysis results are described according to the type of sphalerite.

Response 7: The text describes the types of sphalerite by composition. I tried to redo the table but it turned out very complicated.

Round 2

Reviewer 1 Report

some remarks in attached file
